# Pharmacokinetic Evaluation of a Novel Transdermal Ketoprofen Formulation in Healthy Dogs

**DOI:** 10.3390/pharmaceutics14030646

**Published:** 2022-03-15

**Authors:** Halley Gora Ravuri, Nana Satake, Alexandra Balmanno, Jazmine Skinner, Samantha Kempster, Paul C. Mills

**Affiliations:** 1School of Veterinary Science, The University of Queensland, Gatton, QLD 4343, Australia; h.ravuri@uq.net.au (H.G.R.); nsatake@beckman.com (N.S.); alexandra.lonergan@uq.net.au (A.B.); s.kempster@uq.edu.au (S.K.); 2School of Agriculture and Environment, University of Southern Queensland, Toowoomba, QLD 4350, Australia; jazmine.skinner@usq.edu.au

**Keywords:** pharmacokinetics, dogs, ketoprofen, bioavailability, transdermal

## Abstract

Dogs undergo various surgical procedures such as castration, ovariohysterectomy, and other orthopedic procedures, which are known to cause inflammation and pain. Non-steroidal anti-inflammatory drugs (NSAIDs) are very effective analgesics for alleviating postoperative pain in veterinary medicine. Ketoprofen is currently approved in Australia and the United States for treating different painful conditions in dogs. This study evaluated the pharmacokinetic parameters of ketoprofen after intravenous (IV) and transdermal (TD) administration in healthy dogs. A novel transdermal ketoprofen (TDK) formulation containing 20% ketoprofen, dissolved in a combination of 45:45% isopropanol and Transcutol, along with 10% eucalyptus oil, was developed and evaluated for in vitro dermal permeation using Franz diffusion cells. A crossover study was then conducted to determine the pharmacokinetic parameters of the formulation in six dogs following IV ketoprofen (1 mg/kg) and TDK (10 mg/kg) administration. A liquid chromatography–mass spectrometry (LC-M/MS) method was used to measure plasma concentrations of ketoprofen over time, and a non-compartmental analysis determined the pharmacokinetic parameters. The mean terminal elimination half-life (*T*_½_ h), *AUC*_0-t_ (µg·h/mL), and mean residence time (MRT, h) between IV and TDK groups were 4.69 ± 1.33 and 25.77 ± 22.15 h, 15.75 ± 7.72 and 8.13 ± 4.28 µg·h/mL, and 4.86 ± 1.81 and 41.63 ± 32.33 h, respectively. The calculated bioavailability (*F*%) was ~7%, with a lag time of 30 min to achieve effective plasma concentrations after the application of TDK.

## 1. Introduction

Nonsteroidal anti-inflammatory drugs (NSAIDs) are widely used for treating pain in veterinary medicine. The anti-inflammatory effect of NSAIDs predominantly depends upon their ability to inhibit the activity of cyclooxygenase (COX) isoenzymes, which inhibits the synthesis of prostanoids [1,2], thereby reducing inflammation. Ketoprofen is a propionic acid derivative and is routinely prescribed for its antipyretic, analgesic, and anti-inflammatory properties [3,4]. It is a non-selective COX inhibitor [5] that prevents the synthesis of prostanoids [6], which are major mediators of acute and chronic inflammatory conditions. Ketoprofen may also exhibit centrally-mediated, anti-nociceptive activity [7,8], elicited via supraspinal and spinal receptors [9]. Ketoprofen has provided effective analgesia for clinical conditions in dogs, such as osteoarthritis [10], castration [11], joint pain [12], and various orthopedic procedures [13,14].

Unfortunately, when orally administered, ketoprofen may induce various adverse effects, including gastric ulceration [15], intestinal hemorrhage [16], and kidney damage [17] in dogs. One approach that could possibly reduce these adverse effects of NSAIDs is through transdermal delivery [18]. Several studies have shown the efficacy and safety of transdermally delivered ketoprofen in humans [19,20] and rats [21,22]. Other NSAIDs, such as diclofenac [23], ibuprofen [24], and methyl salicylate [25], have also provided effective analgesia when administered topically in humans.

Only a few studies have reported successful transdermal delivery of NSAIDs in animals, including ketoprofen in cattle [26] and pigs [27], and flunixin meglumine in horses [28], goats [29], and piglets [30]. Some studies targeted local penetration only, including naproxen [31], methyl salicylate [32], meloxicam [33], and celecoxib [34]. Opioids have also been delivered transdermally in dogs, such as fentanyl [35] and buprenorphine [36]. However, NSAIDs target inflammatory-mediated pain, and may even be more useful than opioids for treating inflammatory conditions [37,38]. The current study aims to describe the pharmacokinetics, including bioavailability, of novel transdermal ketoprofen (TDK) formulations in dogs.

## 2. Materials and Methods

### 2.1. In Vitro Experiments

#### 2.1.1. Skin Samples

Skin samples were excised from Greyhound dogs that had already been euthanized for other reasons at the Clinical Studies Centre (CSC), UQ, Gatton campus. The hair on the skin from the thoracic region was clipped using electric clippers (WAHL KM-2 model 1247; blade; Oster A5^®^ Crytotech™, size 40, Unity Agencies, Coorparoo BC, QLD, Australia) and placed in plastic bags on ice. The subcutaneous fascia was trimmed away before skin discs (2 cm in diameter) were excised using a metal wad punch, within 2 h post collection. They were then stored in zip-lock bags at −20 °C. Though the freezing of skin should not alter drug permeability [39], storing dog skin at −20 °C may affect drug permeability after 3 months [40]. Therefore, all in vitro studies that were performed on the skin happened within 2 months. Ethics approval was obtained from the Animal Ethics Committee of the University of Queensland (VETS/ANRFA/516/18) for this study.

#### 2.1.2. Drugs Used for Screening

The drugs selected for this study were non-selective COX inhibitors registered for dogs in Australia: ketoprofen, carprofen, tolfenamic acid, and meloxicam (purchased from BOVA compounding, Shop-4, 304–318 Kingsway Caringbah, 2229, NSW, Australia). Each NSAID was initially screened for its solubility in a large range of vehicles and penetration enhancers (Appendix A). Vehicle combinations dissolving ≥5% [30] of drugs were further modified to determine which excipients, and in which range of concentrations or proportions, dissolved a greater amount of each NSAID. Most of the excipients were mixed with water as a co-solvent and were tested for in vitro drug permeation at a 10% *w/v* concentration (Appendix A). The final candidate formulation (F4), which comprised 10% ketoprofen dissolved in a combination of 45:45% isopropanol and Transcutol (40% in water), along with 10% eucalyptus oil, was selected based on the extent to which it permeated through the dog skin.

#### 2.1.3. In Vitro Experimentation

In vitro studies were performed as described previously [41]. Briefly, skin discs were thawed at room temperature (25 °C) under a fume hood for 1 h. Each formulation was tested in triplicate (three skin discs) per vehicle combination. The thawed skin discs were clamped between the donor and recipient chamber of the Franz diffusion cell apparatus, and ~3.2 mL of 4% bovine serum albumin (BSA; Bovomax; Bovogen Biologicals, Essendon, VIC, Australia) in PBS with 0.1% sodium azide (Sigma-Aldrich Co., St. Louis, MO, USA) was added to the receptor chamber in order to prevent microbial growth. Phosphate-buffered saline (PBS; 1.0 mL) was added to the epidermal surface of the skin (donor chamber) so that it could rehydrate for 1 h, then it was removed and replaced with 1.0 mL of the test formulation (10% *w/v*). The surface area of the exposed stratum corneum was 1.13 cm^2^. The Franz cells were partially immersed in a water bath set at 35 °C and a cover slip was placed over the donor chamber to prevent evaporation. Receptor fluid (200 μL) was collected at regular intervals (0, 0.5, 1, 3, 4, 6, 8, 10 and 12 h) via the sampling port and replaced with the same volume of 4% BSA. All samples were labeled and stored in a −20 °C freezer to be analyzed within 2 months.

### 2.2. Pharmacokinetic Study

#### 2.2.1. Animals

Domestic dogs (*n* = 6) were enrolled in a prospective crossover study. The dogs were aged between 2 and 14 years, and their bodyweight ranged between 23 and 35.4 kg. All of the dogs were clinically healthy, desexed, and were of the following breeds: Labrador (*n* = 1), Greyhound (*n* = 2), cattle dog (*n* = 1), and mixed breed (*n* = 2). All the dogs were owned and housed within the Clinical Studies Centre (CSC) at The University of Queensland (as part of student training for handling animals) and were under veterinarian supervision for the study. Approval for this study was obtained from the UQ Animals Ethics Committee (approval #: SVS/014/19), which follows international, national, and institutional guidelines for the humane treatment of animals.

#### 2.2.2. Drug Treatments and Sample Collection

Dogs were randomly assigned to receive either the IV or TDK formulations first, with at least a 1-week wash-out period (>10 half-lives of ketoprofen, which equated to 9 days once the phlebotomy sites had fully healed) before the alternative treatment was administered. Dogs in the intravenous group (IV) were injected with ketoprofen (1 mg/kg; ilium, ketoprofen injection, 100 mg/mL) into the left cephalic vein via a 22 G × 1.25” catheter as a bolus administration and the catheter was then flushed with saline. Blood samples (3.0 mL) were collected from an 18 G × 3” catheter inserted into the right cephalic vein at 0, 2, 5, 10, 20, 30, 45 min, and then at 1, 2, 3, 4, 6, 8, 12, 24 and 30 h, they were placed into lithium heparin vacutainers (BD, North Ryde, NSW, Australia). For the TDK formulation, ketoprofen (purchased from Bova compounding; Caringbah, NSW, Australia) was used to prepare the topical formulation at 10% and 20% (*w/v*) concentrations. Although both the 10% and 20% ketoprofen concentrations appeared stable, preliminary pilot studies (unpublished results) suggested that insufficient systemic drug concentrations were achieved with the lower active formulation. Therefore, TDK (at 20% *w/v*) was administered at a dose rate of 10 mg/kg (~1–1.5 mL per dog) by parting the hair over the neck region and applying it to the skin surface using a syringe. Blood samples were collected at 0 and 30 min, and then at 1, 2, 3, 4, 6, 8, 12, 24, and 30 h in order to be put into lithium heparin vacutainers (BD, North Ryde, NSW, Australia). The application site was monitored by a veterinarian throughout the sampling phase for any signs of erythema, a skin reaction, or skin irritation as a result of the dog’s actions (i.e., scratching the site), and this was continued on a daily basis for 7 days. The blood samples from both the IV and TDK groups were centrifuged at 3500× *g* for 15 min at 4 °C. All plasma samples were aliquoted into cryovials and stored at −20 °C until further analysis.

### 2.3. Ketoprofen Analysis

Ketoprofen concentrations in receptor fluids and plasma were measured by using a Shimadzu Nexera^TM^ uHPLC connected to a triple quadrupole mass spectrometer, Shimadzu LCMS TQ 8030 (Shimadzu Corporation, Tokyo, Japan). The chromatographic separation of the analyte was performed on a reverse-phase C18 column (Kinetex 2.6 μm, C18, 100 Å; size 50 × 9 × 2.1 mm; Phenomenex Inc., Lane Cove, NSW, Australia), with a SecurityGuard™ Ultra Guard Cartridge (Phenomenex Inc., Lane Cove, NSW, Australia) for chromatographic separation. Mobile phases of 0.1% formic acid in LCMS water and 0.1% formic acid in acetonitrile were used for Pumps A and B, respectively. A binary gradient was used for the elution of the analytes, which increased from 30% of Mobile Phase B to 100% over 3.50 min. Mobile Phase B was then held at 100% until 4.50 min, before being rapidly reduced back to 30% at 4.51 min and maintained at 30% until pump pressures were returned to the stable initial column pressure. The total chromatographic separation was carried out over 7 min. The total flow rate of 400 μL/min between Pumps A and B was maintained throughout chromatographic separation. The column temperature was maintained at 40 °C. The mass spectrometer was operated using an electrospray ion source in the positive ionization mode and was monitored using the developed multiple reaction monitoring (MRM) method for ketoprofen, which gave a result of *m/z* 255.10–105.10, 209.05. Ketoprofen d-3 was used as an internal standard and was monitored at *m/z* 258.0–212, 179. The collision gas pressure was at 230 kPa, and the collision energy was −12 V for both analytes. Dwell time for each transition was set at 100 ms.

A liquid–liquid extraction (LLE) method [42] was modified for extracting ketoprofen from both receptor fluid and plasma samples. Initially, protein precipitation was performed on plasma samples by adding ice-cold acetonitrile (with 0.2% formic acid) at a 1:1 ratio in two rounds, which was vortexed for 30 s each, followed by centrifugation at 20,000× *g* for 15 min. The supernatants from both rounds were collected into a new, labeled Eppendorf tube. Next, 1 mL of methyl tert-butyl ether (MTBE) was added to the samples, which were vortexed for 1 min, followed by centrifugation at 20,000× *g* for 15 min. The collected supernatant was vacuum dried for 2 h at 30 °C using a SpeedVac. The dried samples were reconstituted with 30% acetonitrile and were spiked with a 20 ng/mL concentration of ketoprofen–D3 as an internal standard (IS), which was analyzed on LCMS. The extraction recoveries of ketoprofen and IS were in the range of 85–90% in this study.

Blank dog plasma samples were spiked with ketoprofen over a linear range between 1.5 and 1000 ng/mL as calibration standards and subjected to the same extraction protocol as all other test samples. Quality control samples were prepared in mobile phases at 10 ng/mL for lower quantification, 25 ng/mL for middle, and 100 ng/mL as higher quantification. The LOD was calculated as 1.9 ng/mL and the LOQ was calculated as 5.3 ng/mL. The calibration curve showed a linear response of r^2^ = 0.9991. Calibration standards and quality control samples were analyzed three times, along with all other test samples, at the beginning, middle, and end of the batch run.

### 2.4. Pharmacokinetic Parameters Calculation

The kinetic parameters (Table 1) were calculated using non-compartmental (Trapezoidal) analysis in a PK solver [43]. The maximum plasma concentration (*C*_MAX_) and time to *C*_MAX_ (*T*_MAX_) were directly calculated from the data. The *AUC* vs. time curve (*AUC*_0–∞_) was calculated using the linear trapezoidal rule, and the TDK *F*% was calculated from the ratio of the *AUC*, after TDK and IV administration, indexed to their respective dose:*F* (%) = [(*AUC*_TDK_ × Dose_IV_)/(*AUC*_IV_ × Dose_TDK_)] × 100

### 2.5. Data Analysis

Data were analyzed using GraphPad Prism^®^ (GraphPad Software, La Jolla, CA, USA). The data were tested for normality using D’Agostino–Pearson omnibus K2 normality tests. Most of the data were non-Gaussian, so non-parametric statistical tests were used throughout. The data generated for both groups were compared using Student’s *t*-test at *p* < 0.01 significance.

## 3. Results

### 3.1. In Vitro Screening

From the initial solubility test, ketoprofen, meloxicam, and carprofen were soluble in most of the vehicles at 1% *w/v* concentration. Tolfenamic acid was insoluble in all these vehicles and was not considered further. Meloxicam would not dissolve in any of the formulations at 5%, so was also discarded. Ketoprofen was soluble in many of the solvents at 5 and 10% concentrations, as was carprofen, although to a lesser extent, and only in two vehicles (Appendix A). The screening also showed that alcohols dissolved NSAIDs more effectively at a concentration of 40–50%, while terpenes were more effective at lower concentrations, particularly eucalyptus oil at 10% (Appendix A). An initial screening study was performed with 16 vehicle combinations (Appendix A), and based on the results, the in vitro penetration (Figure 1) of ketoprofen was significantly higher with Formula 4 (F4; *p* < 0.01), compared with all other formulations. This formulation (TDK) exhibited the highest permeation through dog skin and was therefore selected for the in vivo study.

### 3.2. Pharmacokinetic Study

The application of the TDK formulation did not elicit any signs of skin irritation during the 30 hrs study time or when monitored over the following 7 days. There were significant differences (*p* < 0.01) in the mean terminal elimination half-life (*T*_½_ h), *AUC*_0-t_ (µg·h/mL), and mean residence time (MRT, h) between IV (Figure 2) and TDK (Figure 3) groups (4.69 ± 1.33 and 25.77 ± 22.15 h; 15.75 ± 7.72 and 8.13 ± 4.28 µg·h/mL; 4.86 ± 1.81 and 41.63 ± 32.33 h, respectively), and a bioavailability (*F*%) was calculated for a TDK of ~7% (Table 1). However, the TDK formulation was eliminated more slowly than the intravenous dosage, resulting in a significantly higher (*p* < 0.001) MRT for TDK compared with IV.

## 4. Discussion

This is the first study to demonstrate the transdermal penetration of ketoprofen through the skin of healthy dogs. Very few studies have reported transdermal penetration of NSAIDs in dogs, including naproxen in Beagle dogs [31], although the bioavailability was low (~2%), and the calculated half-life differed from the IV administration of the same drug, suggesting a possible ‘flip-flop’ effect [44]. Other studies focused on local or regional delivery. For example, topical administration of meloxicam [33] resulted in clinically significant concentrations of the drug in the synovial fluid under the site of administration only, but low systemic concentrations due to a low bioavailability (~1.05%). Similarly, therapeutic concentrations of methyl salicylate were measured in the synovial fluid and surrounding muscle of the hip when a commercially available human formulation was applied to the skin overlying this joint [32]. A major outcome of the current study was that therapeutic concentrations of ketoprofen were achieved in the systemic circulation, reflecting the higher (~7%) bioavailability. Equally important, is that the formulation was well tolerated, with no evidence of erythema, skin reaction, or irritation as a result of the dog’s actions (i.e., scratching the site), at any stage during the 7 days of observation following application.

Ketoprofen was injected via IV methods in order to permit the calculation of bioavailability and to compare the pharmacokinetics of previous studies. The plasma clearance was rapid (Figure 2), while the terminal elimination *T*½ (4.28 ± 1.6 h) was longer than had previously been reported for racemic ketoprofen—1.36 ± 0.52 h; [45] or 2.57 h for S-(+) enantiomer [46]. However, in a different pharmacokinetic study of S(+) ketoprofen enantiomers [47], the pharmacokinetic parameters *T*½ (4.93 ± 0.69 h), MRT (5.76 ± 0.9 h), and Cl (0.08 ± 0.02 L/h kg) were found to be similar to the current study. The V*d* reported in the present study was similar to results reported by [46], but higher than results published by [45]. Thus, there appears to be some variability in the literature in terms of the pharmacokinetics reported for ketoprofen in dogs following IV administration. However, there were also several similarities to the present study.

The issue of enantioselective disposition kinetics and a chiral inversion was considered when designing this study, but the primary focus was to determine if we could achieve sufficient penetration of ketoprofen through canine skin and attain systemic therapeutic concentrations. Furthermore, for racemic drugs with linear pharmacokinetics and minimal to modest stereoselectivity in their kinetic parameters, stereospecific analytical methods are not warranted [4,48]. Importantly, the chiral conversion will primarily occur once ketoprofen has passed through the skin and reached the liver [49], so it could be expected to affect racemic IV and TDK formulations in a similar manner. There is some information in the literature to suggest that different routes of administration may affect the degree of the chiral conversion of ketoprofen in calves [4]. However, these same authors reported that ‘the plasma concentrations of both enantiomers are similar in calves despite the occurrence of chiral conversion, which corresponds to the observations in goats and humans’. Therefore, future studies are warranted to determine if the transdermal permeability differs between enantiomers, as this was not within the scope of the current study.

A major outcome of the TDK formulation was a very short lag time to reach systemic circulation (~30 min), and a rapid attainment of therapeutic concentrations (>2 ng/mL, [50]). Fentanyl is a commonly used opioid analgesic in dogs, but the lag time can be prolonged for up to 12 h using a patch, or 2–4 h with a novel gel [35], whereas buprenorphine has a lag time of >24 h when applied to dogs as a patch [51]. The *F*% of the TDK was 7.16% and the median C_MAX_ was 0.54 µg/mL, which was higher than reported in topical ketoprofen studies in humans. For example, Shah et al. [52] reported an *F*% of 1% and a C_MAX_ ranging between 0.08 and 0.119 µg/mL, whereas another study [53] reported a relative bioavailability of 0.48% after the application of a 20% ketoprofen gel.

When compared with previously published oral ketoprofen pharmacokinetic studies in dogs, the observed *C*_MAX_ for TDK (0.52 ± 0.28 µg/mL) was significantly lower than the *C*_MAX_ of oral ketoprofen (2.02 ± 0.41 µg/mL [54]; 1.13 ± 0.34 µg/mL [47] and 4.91 ± 0.76 µg/mL [46]). It is particularly interesting that the orally administered ketoprofen is rapidly cleared from systemic circulation and is not detectable beyond 6–8 h. The TDK administration resulted in plasma ketoprofen concentrations that were still above the effective plasma concentration, i.e., >2 ng/mL [50] for 24 h after application, which has important implications for efficacy and safety. The recommended oral dose rate in dogs is 1–2 mg/kg PO [55], but this is only recommended for 3 days to avoid toxic adverse effects. However, adverse effects related to ketoprofen are believed to be dose-dependent [56], and indeed, oral dose rates of 0.5 mg/kg [10] or 0.25 mg/kg [12] have effectively controlled pain over a prolonged period (28–30 days). It is therefore feasible that the TDK formulation may provide a longer duration of analgesic efficacy in the dog with a potentially reduced incidence of adverse effects. The prolonged terminal half-life (*T*½) observed for TDK has also been reported for other transdermal NSAIDs in dogs, such as meloxicam (36.6 ± 6.96 h) [33] and naproxen (61.2 + 12.8 h) [31]. This may reflect entrapment of the drug in the skin layers and sustained slow release [33], contributing to a ‘flip-flop’ effect as the absorption rate constant exceeds the elimination rate constant [44].

## 5. Conclusions

Topical delivery of ketoprofen resulted in therapeutic systemic drug concentrations with a short lag time. Prolonged plasma ketoprofen concentrations, compared with oral dose rates, suggest a longer period of efficacy, but with a possible lower incidence of adverse effects. Further studies are required to determine its efficacy and to investigate its safety in dogs with inflammatory conditions, and to determine the use of TDK as a pre-emptive analgesic prior to surgery. The developed novel transdermal formulation could be further evaluated for its efficacy under a clinical scenario.

## Figures and Tables

**Figure 1 pharmaceutics-14-00646-f001:**
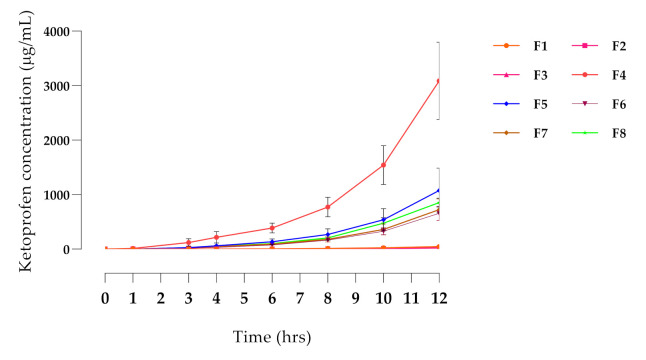
In vitro permeation profile of ketoprofen through excised dog skin. Formulations (F1–F8) were investigated in terms of penetration of the active drug. Each data point represents an average value of sample concentration from three replicates (*n* = 3).

**Figure 2 pharmaceutics-14-00646-f002:**
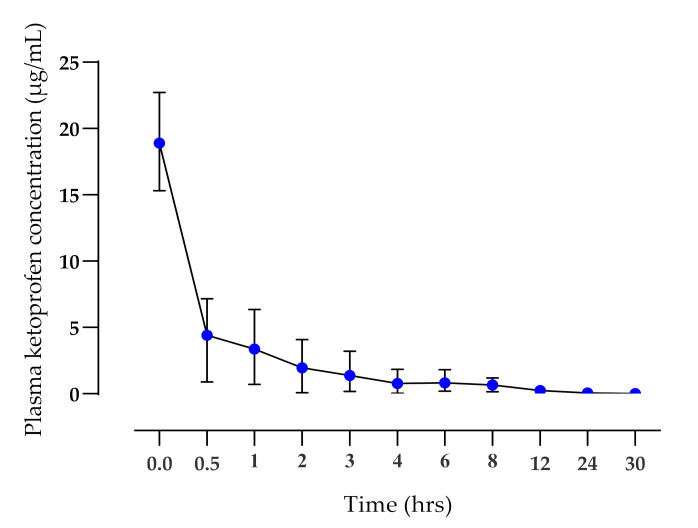
Plasma ketoprofen concentrations (mean ± SD) after IV administration (1 mg/kg) to healthy dogs (*n* = 6).

**Figure 3 pharmaceutics-14-00646-f003:**
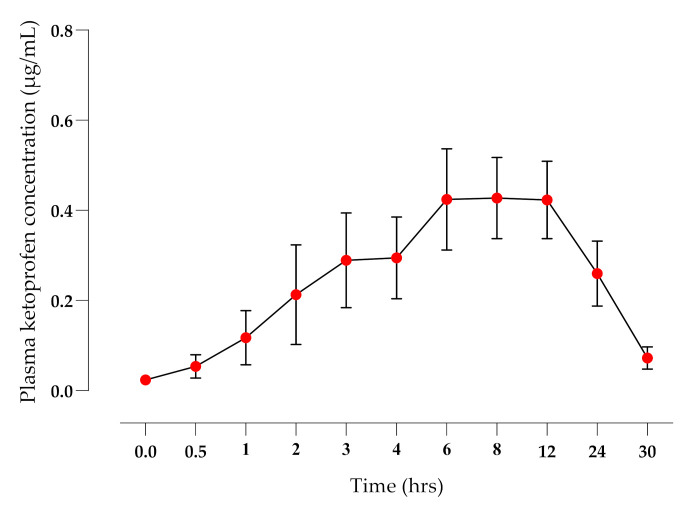
Plasma ketoprofen concentrations (mean ± SD) after TDK application (10 mg/kg) to healthy dogs (*n* = 6).

**Table 1 pharmaceutics-14-00646-t001:** Pharmacokinetic parameters were calculated for ketoprofen after a single IV bolus (1 mg/kg) and after a single topical application (10 mg/kg) of TDK formulation in six healthy dogs.

Parameter	Unit	IV Route (Mean ± SD)	TD Route (Mean ± SD)
Lambda_z (λ_z_)	L/h	0.16 ± 0.04	0.05 ± 0.06 *
*T*½	h	4.69 ± 1.33	25.77 ± 22.14 *
*T* _MAX_	h	0.03 ± 0.0	9.14 ± 3.98 *
C_MAX_	µg/mL	16.43 ± 5.32	0.52 ± 0.28 *
*AUC* _0-t_	µg·h/mL	15.75 ± 7.72	8.13 ± 4.28 *
*AUC* _0-_ _∞_	µg·h/mL	16.71 ± 8.46	19.61 ± 16.94
*MRT*	h	4.86 ± 1.81	41.63 ± 32.33 *
*Vd*	L/kg	0.48 ± 0.18	21.69 ± 15.64 *
*Cl*	L/h kg	0.08 ± 0.04	0.77 ± 0.43 *
*F* (bioavailability)	%	NA	7.16 ± 7.08

* Significant at *p* < 0.01. Abbreviations: IV, intravenous; TDK, transdermal ketoprofen; *AUC*_0-t_, area under the curve; *AUC*_0-∞_, area under the curve calculated from the time of dosing to infinity; *MRT*, mean residence time of the drug in the body; *C*_MAX_, maximal plasma concentration; *T*_MAX_, time of C_MAX_; *Vd*, the volume of distribution; *Cl*, plasma clearance; *T*½, terminal half-life; Lambda_z (λ_z_), elimination rate constant; *F*, TDK bioavailability; N/A, not applicable.

## Data Availability

All data is available in the Appendix A.

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
