# Peer review of "Pharmacokinetic Evaluation of a Novel Transdermal Ketoprofen Formulation in Healthy Dogs"

_pharmaceutics, 2022, doi:10.3390/pharmaceutics14030646_

Round 1

Reviewer 1 Report

Minor suggestions:

  • ''developed using in-vitro Franz cell screening'': Please change to- Developed and evaluated for in-vitro dermal permeation using Franz diffusion cellls.
  • The developed novel transdermal formulation could further be evaluated for its effi-25 cacy under clinical scenario. Shift this line from abstract to conclusion.
  • Authors have not mentioned ‘Materials’ as a separate section
  • Authors have mentioned formulation composition only in abstract section. How the formulation was prepared should also come in methods section. 45% of transcutol, 45% of IPA and 10% Eucalyptus oil will be very irritating to skin. Authors should have used only minimum sufficient quantities of these 3 ingredients, just to dissolve Ketoprofen and rest of the vehicle should be water. In my opinion this formulation will not qualify the permissible limits of these ingredients in topical / transdermal products. Please check guidelines and mention the same in the manuscript. Otherwise use of such irritant solvents is not justifiable.
  • Symbol of degree is not correctly typed.
  • Abstract mentions that formulation contains 20% ketoprofen. However, section 2.1.3 has used 10%. Discrepancy.
  • 2.2. Why TDK was applied 10 mg /kg where as IV dose was 1 mg/kg?
  • In conclusion authors have compared TDK data with oral. However, PK studies were carried out IV
  • TDK T half was more than 24 h. So it does not match with the statement ‘’ 1-week wash-out period (> 10 half-lives of ketoprofen’’

Reviewer 2 Report

This is an interesting work on an area with little information, so teh results are of value for other researchers.

I have the following comments.

  1. in line 55: please indicate what the abreviation TDK stands for 
  2. in line 59. Please edit so it is clear that the skin was excised after euthanasia. The current text suggest it was excised from dogs intended for euthanasia which could be understood before the animal was dead. And perhaps this information should be closer to that on the approval by the ethics committee that was probably more concerned with the animals than with the effect of freezing the skin
  3. Line 83: would you clarify whether the 1 mL PBS was used to equilibrate the subdermal or the epidermal side of the skin or the whole skin and so whether this took place in a tube or somewhere else
  4. line 116; this is at the journal discretion but "@" should be "at", perhaps?
  5. With regards to the transdermal administration to animals in vivo, the formulation used, the area of application and the dose applied per unit area should be provided. Also was any hair clipping or other procedure in addition to parting the hair  done before application? WAs the formulation used any of the F1 to F8 in Fig.1?
  6. To avoid confusion, please indicate in Figures such as Fig. 1 (or any other relevant) that the data refers to excised skin or to in vitro permeation tests. In addition the mean value should be accompanied by a measure of variability, standard deviation preferably, even acknowledging that calculation with n=3 is somehow hindered but just to give the reader an idea of the variability observed
  7. in line 196. it is said that no signs of irritation were observed. However, the methods used for determining this are not detail in methods
  8. Line 198 and related. Unless the authors are claiming non-linear kinetics, the elimination half-life of a drug is constant. Hence, the elimination rate constant is that one determined in the IV administration. The TD data is consistent with the much slower absorption across the skin which leads to the classic flip-flop kinetics situation which is very interesting. However, referring to this value as the elimination half-life is not correct, it should be referred to as the terminal phase half-life and it corresponds probably to the half-life of absorption as this process becomes the rate limiting step. This effect is very often seen with human and transdermal patches and there are references available to compare with this data. As the authors refer to flip-flop kinetics on and use the better term "terminal half-life" later on, this should be obvious to correct here
  9. The units for the parameters in Table 1 are not consistent with the usual units for these parameters. I suggest transforming units in the data enter to the so they are consistent before the modelling/data analysis and the parameters are obtained with useful units and the values can be used by other researchers for comparison and further research. I understand they still be normalized by body weight,  but the volumes should be volume or volume/kg (if normalized) and clearance in volume/time or( volume / time)/kg if normalized. But it is not OK to have several mass units like mg and µg mixed when this can be easily sorted.
  10. in line 207: the legend of the picture there is an "area" that seems to have a different font then the rest of the text
  11.  The X axes on Figs 2-3 (time) could have more divisions (vertical marks)  to aid appreciation of the time for different samples. For example, in Fig.8 there are 8 different concentrations values between time 0 and 10 so the X axes should facilitate assigning the time to these concentrations, for example have 5 also indicated and little vertical marks in between.
  12.  In line 238 the authors compare the clearance measured with that in other studies. However, how was it possible to compare the 0.08 ± 0.04 here measured (Table 1)  in (mg/kg)/(μg/mL)/h with the value Cl (0.08 ± 0.02 mL/kg/hr)  in line 238? The units are not consistent see point 9 above). The same comment probably concerns the lines 239-240 regarding the volume of distribution which again has inadequate units in Table 1. The values reported in other work for V could be provide so the reader can make their own opinion on the similarity / lack of.  
  13. Lines 258-266. Time to steady state will differ for different drugs and species which is well-known and obvious, so what is the point here that the authors are trying to make?
    Also I find biased that the authors compare their data to that of drugs that permeate more slowly the skin but not to drugs that go very fast  like nicotine. So again, what is the purpose of this comparison? Are the authors suggesting that transdermal ketoprofen should be used instead of fentanyl or buprenorphine, indication for NSAIDs and opioid drugs are different. If they want to refer to the field in general then a review should be used rather than cherry pick examples.
  14. In Table 2 of supplementary data: it is unclear what the second solvent(s) is in the first 6 systems.
  15. What is the importance of the appendegeal path in the dog? How does this impact in the comparison to human transdermal fluxes and and perhaps between different dog breeds?
  16. Finally, the topical/transdermal use of ketoprofen in humans has been hindered by photosensitivity reactions. What could be the impact of these in dogs? 
  17. Please have a good read of the whole text, there are several sentences that need improved editing

Round 2

Reviewer 2 Report

Many thanks for the corrections done that will make your work clearer to other authors. I have to minor corrections: 

In line 209. The term elimination half-life instead of terminal half-life is still used. This shows now correctly in the table

In Figure 1, in the Y axes, the symbol for micrograms seems a bit blurry, It could be my laptop but please ensure that it is clearer in the final version
